# Drawbacks of Olanzapine Therapy: An Emphasis on Its Metabolic Effects and Discontinuation

**DOI:** 10.3390/jcm14228125

**Published:** 2025-11-17

**Authors:** Ramadhan Oruch, Hussein Abdullah Rajab, Mahmoud Abdalla Elderbi, Ian F. Pryme, Ole B. Fasmer, Anders Lund

**Affiliations:** 1Department of Biochemistry and Molecular Biology, School of Medicine, Najran University, Najran 66462, Saudi Arabia; 2Department of Clinical Medicine, Section of Endocrinology, School of Medicine, Najran University, Najran 66462, Saudi Arabia; 3Department of Pharmacology and Toxicology, School of Medicine-Almarj, Benghazi University, Benghazi 18251, Libya; 4Department of Biomedicine, School of Medicine, University of Bergen, 5009 Bergen, Norway; 5Department of Clinical Medicine, Section of Psychiatry, University of Bergen, 5009 Bergen, Norway

**Keywords:** intoxication, weight gain, insulin resistance, 5-HTR2C receptor, htr2c, free radicals, mitochondrial dysfunction, serotonin signaling, drug interactions

## Abstract

Radical drug therapy for schizophrenia is usually hard to achieve with one currently available antipsychotic agent. Indeed, it is the negative symptoms of this morbidity that are a dilemma to neutralize. Most of the first-generation agents can deal with the positive symptoms of the disease to a convincing degree, but not with its negative symptoms. The creation of so-called second-generation agents aimed to treat the negative symptoms, as these invisible barriers are the real reasons that isolate psychotic individuals and hinder their integration into society. Unfortunately, these newly designed drugs, including OLZ, turned out to induce different categories of undesired effects; the most embarrassing among them are the metabolic drawbacks, such as insulin resistance, weight gain, and other subcategories of metabolic consequences. Antagonism induced at certain receptors, particularly 5-HT2C and histamine H1 receptors, is implicated particularly in these metabolic adverse effects. The choice of antipsychotics (APCs) should be tailored separately for each case, as each patient responds variably to each neuroleptic. This possibility exists due to the abundant alternatives within the currently available APC medications. This work aims to discuss the reasons behind these undesired metabolic effects, how to deal with them, how to choose the appropriate agent for each psychotic case, and how to manage intoxication using olanzapine. To address these inquiries, we carefully selected 154 relevant studies, including robust meta-analyses, from the past 20 years and analyzed them in this work.

## 1. Introduction

Psychosis, depression, and anxiety neurosis disorders are the three main classes of mental diseases that can affect humans. Schizophrenia is the major subclass of psychosis. Depression, which includes both major depressive disorder and bipolar disorders, is another major subclass of mental diseases, and its incidence in different societies around the globe is higher than that of psychosis. Anxiety and insomnia constitute the third subclass of neuropsychiatric ailments that psychiatrists/neurologists encounter and treat in their clinics. According to available statistical data, more than 1/3 of individuals in any society at some point in their lives manifest clinical criteria sufficient for one or another form of psychiatric condition to be diagnosed [1]. In the United States, about 46% of the population at some point qualifies for a mental illness diagnosis [2].

Among the relatively newly designed APC agents is olanzapine (OLZ), which we are going to tackle in this work. In addition to its major therapeutic use as an APC, OLZ has other uses that we will mention briefly.


**Antipsychotics and classification:**


Antipsychotic agents constitute a class of medications that is primarily used to treat schizophrenia. Together or as an adjunct to mood stabilizers, they are also used in the management of bipolar disorders. Dependent on the dose being therapeutically used, they also have many off-label uses, as is the case with quetiapine and OLZ. Antipsychotics are traditionally classified into first and second generation agents [3]. Being non-selective in their action, first generation agents cause a wide spectrum of undesired effects, some of which are associated with the drugs themselves, and some result from their interactions with other psychotropics and many other drugs that are used to treat different somatic morbidities. Moreover, many first generation agents are impotent in their therapeutic actions [4]. Recall that some of the side effects exerted by these agents, idiosyncratically, can be devastating, including neuroleptic malignant syndrome (NMS), serotonin syndrome (SS), blood dyscrasia, myocarditis, and others. All these factors motivated drug industry corporations, via the drug designers, to manufacture more selective and potent agents to deal with the dilemma of psychosis. These firms have made great efforts to create modern medicines that are therapeutically highly effective and have fewer side effects. They have recruited their scientists to manufacture what is now known as second generation agents. Unfortunately, it turned out that these newly prepared agents can also exert the same side effects as first generation agents (probably to a lesser extent). Moreover, they also have their own drawbacks, particularly of a metabolic type, especially insulin resistance, and the consequences of insulin resistance include type 2 diabetes, which is an early phase in the spectrum of metabolic syndrome. The reason these adverse effects occur as a result of OLZ, similar to other currently available APC agents, is that it is a non-selective agent although being potent [5,6]. OLZ, in addition to D2 receptors’ antagonism, also antagonizes many receptors such as histamine H1 and serotonin HTR2C, which are believed to be the reason behind the metabolic drawbacks OLZ causes; that is, OLZ exerts multi-receptor antagonism characteristics.


**Olanzapine:**


OLZ is a thienobenzodiazepine derivative (water-insoluble) and a second generation APC agent developed by Eli Lilly in 1971; it was patented in the US in 1996. The chemical formula of the agent is C_17_H_20_N_4_S, which has a molecular mass of 312.439 g/mole, and the PubChem CID is 4585. OLZ is well-known under the trade name Zyprexa. It was basically developed to treat schizophrenia and bipolar affective disorders [7]. OLZ is more efficacious than a number of other second generation APCs in the long run, but its efficacy must be weighed against its adverse effect profile [8]. The agent is administered both orally and parenterally (intramuscular IM), which is attributed to the fact that the drug is practically water-insoluble (see the chemical structure) (Figure 1).


**Pharmacopeia of olanzapine:**
**For oral use (including orally disintegrating tablets),** 2.5 mg, 7.5 mg, 10 mg, 15 mg, and 20 mg tablets are available.**For parenteral use,** short-acting intramuscular (IM) OLZ can be used for the treatment of agitation episodes of acute schizophrenia and is an appropriate choice to start with; once the acute psychosis is controlled, one can switch to an oral preparation as maintenance therapy. Long-acting parenteral (IM) maintenance therapy can be used as OLZ pamoate is available and indicated, provided that circumstances are not against such a procedure; this is the new trend of deinstitutionalization of patients with chronic mental diseases.



**Dedicated uses of olanzapine and the recommended doses:**



**Schizophrenia:**
Start initially at a daily dose of 5–10 mg tablets; if necessary, it can be titrated upward with increments of 5 mg/day at intervals of more than a week to reach the maintenance dose of 20 mg daily but not more.Based on oral dose, recommended (IM) doses are as follows:
Daily oral dose of 10 mg is equivalent to 210 mg of IM dose every 2 weeks, or 405 mg of IM every 4 weeks (for first 8 weeks), so 150 mg every 2 weeks or 300 mg every 4 weeks.Daily oral dose of 15 mg is equivalent to 300 mg of IM dose (for the first 8 weeks) every 2 weeks, followed by 210 mg IM (every 2 weeks) or 405 mg IM (every 4 weeks).Daily oral dose of 20 mg is equivalent to 300 mg IM (every 2 weeks); for the first 8 weeks, if titration is optimal against therapeutic effects, the maintenance dose is 300 mg of IM (every 2 weeks). The data regarding OLZ doses dedicated to schizophrenia mentioned above is obtained from [9].



**Mania of bipolar I disorder:**


OLZ is indicated for the treatment of such a condition and also as a maintenance therapy of bipolar I disorder, especially when this is associated with manic or mixed episodes. It is sometimes used as an adjunct to lithium or valproate to treat manic and/or stabilize mixed episodes associated with bipolar I disorder [10].

OLZ is best to be prescribed as a monotherapy, as combination therapy of psychotropics is not recommended because of the possible augmentation of side effect panorama and the avoidance of drug interaction complications. Unfortunately, this practice is not always possible as individual patients respond variably to almost all APC agents; moreover, to obtain the desired therapeutic effect, one has to combine other agents.

**As monotherapy,** take 10–15 mg tablets orally on a daily basis to start with; if needed, titrate upward by 5 mg daily at intervals of 24 h.

**As an add-on** to lithium or valproate, take 10 mg orally on a daily basis to start with.

**As a maintenance therapy,** take 5–20 mg tablets daily, and this dose must not exceed 20 mg.

**To treat agitation associated with schizophrenia and mania of bipolar type I**, take short-acting 2.5–10 mg (IM) to start with and, on-demand, an additional 10 mg (IM) at intervals of 2 h after the initial dose, and if necessary, another 10 mg 4 h after the second dose. This practice is dependent on the clinical outcomes of the individual case in question.

**The treatment of depression of bipolar I disorder** is indicated as a combination therapy with fluoxetine (not more than 75 mg daily), which is usually adjusted at daily doses of 5–12.5 mg in the evening.


**Off-label uses of olanzapine:**
Eating disorders, generalized anxiety disorder, panic disorder, delusional parasitosis, and post-traumatic stress. The use of OLZ in these disorders has not been evaluated rigorously enough.This agent has been used for Tourette syndrome and stuttering [11].Attention-deficient hyperactivity, aggressiveness, and repetitive behavior of autism [12].In insomnia, the effect is comparable to quetiapine and lurasidone [13]. In some cases, the sedation caused by OLZ impairs the ability of individuals to wake up at a steady, consistent time every day. Long-term studies of the safety of OLZ in treating insomnia are still to be conducted.It can be taken as an antiemetic in individuals after receiving anticancer agents because of the high risk of vomiting. As one can see, these off-label uses of the agent are an advantage for this drug, although nobody knows exactly how this agent cures all these different morbidities.



**Olanzapine in pregnancy and lactation:**


Teratogenic effects of OLZ have been well-documented in animal studies [14,15]. In humans, the data concerning teratogenicity is indeed scarce apart from a few case reports [16]. Therapy with OLZ during pregnancy is not absolutely contraindicated. Most of the documented teratogenicity of OLZ is related to its metabolic effects. Overweight and obesity in pregnant women are risk factors for congenital malformations in their offspring. Results from a comprehensive meta-analysis support this claim. Specifically, women with a baseline body mass index (BMI > 30 kg/m^2^) are particularly vulnerable to giving birth to offspring with neural tube defects [17]. OLZ can complicate this issue because it has the highest placental exposure among atypical APCs [16] and can induce obesity. This is attributed to its lipid solubility. It easily crosses the placental barrier, leading to weight gain and increasing the risk of neural tube defects, such as spina bifida, in the offspring of OLZ-treated women [18]. OLZ is secreted into breast milk in very small amounts, so breastfeeding can generally be recommended if there are no alternatives to OLZ [19].


**Olanzapine in older adults and adolescents:**


This agent (and risperidone) should not be prescribed for old patients with dementia due to the increased risk of stroke according to the UK Committee on the Safety Medicines (CSM). In the US, the agent has a warning for increased death risk in older individuals. The agent is not recommended for dementia-related psychosis [20]. Older patients are more prone to experience obesity with this agent compared to risperidone or aripiprazole [21]. The agent causes EPS reactions of all categories in older patients, but dystonia is more commonly seen in young adults. In older individuals and those who suffer hepatic impairment, the therapeutic OLZ dose should be under the standard value.

In adolescents with precautions, OLZ may be indicated for the treatment of schizophrenia and manic or mixed episodes associated with bipolar I disorder in individuals aged 13–17 years [22]. OLZ and other agents of the second generation APCs, such as risperidone and aripiprazole, are also used to treat irritability, hyperactivity, inappropriate speech, social withdrawal, and stereotypy in autism spectrum disorders (ASDs). The common side effects of these three agents are excessive sedation, sleepiness, and weight gain. These side effects are unfortunately more encountered with OLZ than the other second generation agents [23,24]. In all cases, the second generation APC agents, including OLZ, have substantial adverse effect profiles, and each drug has its specific parameters which should be taken in consideration when designing treatment decision [25].

**Metabolism:** After the first pass and reaching the liver, OLZ binds to proteins by 90%. Almost 40% of the oral dose of the agent on average is removed by the hepatic first pass [26]. Peak plasma time after absorption is 6 h (oral route), 15–45 min (short-acting I.M), and 7 days (extended-release IM). The half-life (t/2) of the agent is about 21–54 h for short-acting immediate release preparations and 30 days for extended-release.

The clearance of OLZ varies according to sex and race. Women have an almost 25% lower clearance rate than men, and Afro-Americans have a clearance rate of 26% higher than Caucasians and Asian (Chinese or Japanese) individuals [27]. The drug is metabolized by the cytochrome P450 (CYP) system mainly via isozyme 1A2 (CYP_1_A_2_) and to a lesser degree by the isozyme CYP_2_D_6_. Metabolites of OLZ are inactive pharmacologically; they are metabolized by glucuronidation (water-soluble metabolites, after association with glucuronic acid) and thus mainly excreted via the renal route (57%) and to a lesser extent (30%) eliminated via feces. OLZ causes dysbiosis in the gut microbiota that might also give a clue about the reasons behind OLZ-induced weight [28]. Monitoring of OLZ plasma levels is not reliable because it is not absolutely linear with the therapeutic dose, but is requested to find out the patient’s adherence to the drug therapy as it helps in the tapering of OLZ [29].

**Side effects of the agent:** Performing a rapid check of the literature [30,31], one can realize that OLZ might cause any of the known side effects that typical (first generation) and/or atypical (second generation) APCs can exhibit, in addition to those of benzodiazepine (diazepam) potentiation. The latterly mentioned assertion is because of the lipophilicity of these two agents. OLZ can readily cross cellular biomembranes, including both cytoplasmic and the membranes of intracellular organelles such as the nucleus and mitochondria. Because of this characteristic, it reaches the receptor and binds to certain pockets within the receptor structure. A degree of lipophilicity is essential for a drug to target the receptor within the neurons of the central nervous system. Lipophilicity plays a crucial role in the interplay between any lipophilic drug and the lipophilic moiety of the receptor’s amino acid residues. The lipophilicity must be good enough to create an affinity to attach to these binding sites of the receptor. Medicinal chemists take this issue into consideration when they design new psychotropics or other agents to ensure a certain balance between lipophilicity and hydrophilicity. If the drug is extremely lipophilic, it accumulates nonspecifically in the adipose tissue and other lipids in the body, causing potent toxicity and reduced concentration at the targets that the drug is designed to target [32]. Another pharmacodynamic parameter that has to be mentioned in this context is the Ki value of psychotropic agents. Ki value is the inhibitor constant of a drug; it measures the affinity of the drug to its target receptor or the enzyme that the drug regulates its catalytic function. The lower the value is, the stronger is the binding affinity is.

Both agents (OLZ and diazepam) cause sedation, although by different cellular mechanisms. As a chemical structure, the agent resembles benzodiazepines (see Figure 1). The major possible effects OLZ can cause include extrapyramidal symptoms (EPSs), NMS, blood dyscrasia, chest infections, CNS problems, mental side effects, autonomic nervous system side effects, balance problems, cardiovascular morbidities, musculoskeletal symptoms, urinary system problems, gastrointestinal discomforts, genito-sexual problems (both males and females), dermatological, and others to variable extents [33]. As one can see here, although in different degrees, OLZ can cause side effects in all systems of the human body without exception.


**Obsessive–compulsive disorder (OCD) and paradoxical drawbacks:**


Many psychotropics used to treat various mental illnesses can induce or even cause OCD in individuals undergoing OLZ therapy [34,35,36]. As shown, treating one mental disorder can lead to the development of another psychological issue. Additionally, OLZ itself can sometimes produce other side effects, though in a small subset of individuals. These effects include changes in thoughts, agitation, irritability, personality and behavior abnormalities, and even hallucinations (the intended target of the drug); the most severe of these are depression and intrusive thoughts about suicide [26].


**Post-injection delirium syndrome:**


The condition is one of the potential adverse effects that is unique for OLZ among other second generation long-acting injectable APC agents. The syndrome is also known as post-injection delirium/sedation syndrome (abbreviated as PDSS) as it manifests the clinical picture of both delirium and sedation. This secondary effect exhibited by OLZ pamoate is not seen with other similar injectable long-acting preparations of APC agents of the same class, such as paliperidone palmitate. The syndrome includes the symptoms and signs of delirium (uncoordinated movements, difficulty of speech, confusion) and sedation. It is believed that this pharmacodynamic effect occurs as a result of the mishap of accidental intravenous administration of OLZ pamoate, which is designated for deep intramuscular use as a long-acting medication, as a measure to help deinstitutionalized schizophrenic individuals. The incidence of the syndrome is indeed very low, estimated as 0.07% of parenteral administrations [37].

Side effects of OLZ are not only confined to the drug’s direct side effects, but also to its interactions with other drugs acting on the CNS, including other psychotropics (antipsychotics, antidepressants, and others) and even other groups of drugs devoted to treating somatic diseases. The registered and known drug–drug interactions with OLZ are about 592 (drugs.com). It is worth mentioning that OLZ can also cause an undesired effect, which sort of overlaps with NMS and SS [38]—a condition that can create a diagnostic challenge.

**Discontinuation:** As is the case with other psychotropics, gradual dose tapering is also indicated with this APC agent. This is necessary to avoid acute withdrawal syndrome and rapid relapse of psychosis [39]. The risk of relapse on neuroleptics can be minimized via a planned gradual tapering, which is attributed to the fact that adaptation to APC agents can last for months or years after the cessation of such agents. Dopaminergic (especially D2 receptors) hypersensitivity (DSS) related to the administration of APC agents results in upregulation of dopamine receptors, which is the proposed reason behind tardive symptoms of abrupt discontinuation of APC and the relapse of symptoms. Therefore, it will be wise to taper APCs, including OLZ, gradually over months or even years [40]. The chemical structure of the agent is very similar to benzodiazepines (see Figure 1), which is a fact that, although partially, may explain why these two agents potentiate the pharmacodynamic effects of each other (both action and side effects) in such a way that they induce deeper sedation and hypotension [41].


**Dopamine supersensitivity syndrome (DSS):**


DSS denotes a condition where dopamine receptors of the brain’s striatal neurons become more sensitive to the neurotransmitter dopamine [42]. This phenomenon leads to amplified biochemical, physiological, and behavioral responses. Long-term OLZ use can trigger the condition as a compensatory upregulation of dopamine receptor expression. The catastrophic consequences of this situation is that it can potentially lead to the development of tardive dyskinesia or, in some cases, treatment-resistant schizophrenia [43]. DSS is more liable to ensue in cases of APC polypharmacy. It can happen with any of first or second generation APC. The most known dopamine antagonist APCs that can cause the situation are haloperidol, OLZ, risperidone, and quetiapine, but it is not limited to these drugs. On the other hand, partial dopamine agonist drugs, such as aripiprazole, can reduce or even prevent DSS [44].

Lipid-soluble long-acting injectable (LAI) APC preparations such as haloperidol decanoate, paliperidone palmitate, LAI risperidone, and LAI olanzapine, which are designed to reduce hospitalization periods of patients, may cause DSS [45], although the issue is perplexing.


**Titration of therapeutic serum concentration of olanzapine:**


It is generally accepted that OLZ is a safe drug provided that the therapeutic serum level of the agent is titrated carefully against its therapeutic effects [46]. Therapeutic drug monitoring (TDM) of OLZ showed that the optimal therapeutic concentration lies between 20 and 40 ng/mL. At this concentration, the patients exhibit the problem of weight gain as an adverse effect. Adverse effects that are observed at concentration levels above 80 ng/mL were of a motoric type (extrapyramidal) and cholinergic. At serum concentrations of less than 20 ng/mL, positive symptoms of psychosis can relapse [47]. As one can see, weight gain is not related to the therapeutic dose. There is a strong linear relation between plasma concentration and clinical response. OLZ dose should be adjusted based on plasma concentrations, particularly in the context of high individual variability [48]. Although dose and serum concentration are linearly correlated, therapeutic doses may result in great variability in serum levels; and the relation between OLZ serum value and its therapeutic effects is low [49] because OLZ can interact with many different classes of co-administered or adjunct drugs. These drugs either increase or decrease olanzapine’s serum levels, thus augmenting or decreasing its therapeutic potency. For example, smoking and carbamazepine decrease OLZ serum levels by inducing the cytochrome P450 via mainly its isozyme (CYP_1_A_2_), thus decreasing OLZ concentration. On the other hand, fluvoxamine (an SSRI antidepressant) inhibits CYP_1_A_2_, so it increases the agent’s concentration. The other P450 cytochrome isozyme, namely CYP_2_D_6_, which is also a clinically relevant one, is inhibited by other factors such as female gender and/or senility [50].


**Management of olanzapine intoxication:**


Serum concentrations of above 80 ng/mL are considered the threshold for the occurrence of toxic effects of the drug, although research studies showed that lethal plasma levels are much higher than this. The management of intoxication generally includes the following broad lines:

First, neuroleptic and SS should be ruled out. Fatalities have been documented with plasma concentrations greater than 1000 ng/mL. Bear in mind that to date, there exists no effective specific antidote to revert intoxication caused by OLZ. This is a disadvantage for OLZ [30]. Clinical signs and symptoms mimic those of poisoning with opioids. This includes agitation, tachycardia, dysarthria, and a decrease in the level of consciousness, which terminates with coma. Suicide or attempts to suicide with this agent, if it happens, would be attributed to dose-related heavy intoxications or the heavy combination of the drug with other agents, such as alcohol and opiates. In this sense, the agent is safe provided that its serum levels are titrated correctly according to the therapeutic effects that are tailored individually for each patient.

Treatment of an overdose is performed mainly by supportive measures in an intensive care unit. The steps include endotracheal intubation, intravenous fluids, observation of cardiopulmonary signs, and mental status [51]. It is worth mentioning that the outcome of poisoning with OLZ is more dangerous than that caused by quetiapine and aripiprazole [52]. This is attributed to the prolonged clearance of the drug [40].


**Monitoring of the metabolic parameters when deciding therapy with olanzapine:**


When therapy with OLZ has been decided, caregivers have to start with checking the following metabolic parameters: baseline weight, BMI (weight in Kg/height in m^2^), blood pressure, fasting blood sugar, and fasting serum lipids (after the patient has fasted for at least 12 h). Fasting lipids should include total cholesterol, LDL, HDL, and triglycerides after an overnight fast. After the initial baseline checking, these parameters should be checked on with three-month intervals and annually for the first year. Weight and BMI should be checked more frequently during the first few months. Those who are at higher risk and whose initial results are abnormal should be monitored more frequently [53]. Waist circumference is also pivotal to check as it reflects internal, or what is called visceral, obesity, which is an important sign in insulin resistance (diabetes type 2). Blood pressure is measured in terms of both systolic and diastolic pressure while the patient is sitting. Ongoing monitoring of the first year should include monthly measuring of weight and BMI (for the first three months) [47,54]. Other general parameters could also be added, such as liver function test, prolactin levels, full blood count, and EKG. The most important parameters are demonstrated in Table 1. The interval of checking should be shorter whenever the value of blood sugar increases or other parameters are changed metabolically. OLZ-induced weight gain is reversible on stopping or switching to other drugs [55,56]. Alcohol should be avoided at the start of OLZ treatment as this causes augmented drowsiness and affects alertness and concentration. Alcoholic beverages also increase body weight, which is an unwanted parameter for those who are on OLZ therapy. Moreover, it can cause agitation, aggression, and amnesia. When another APC is added as an adjunct to OLZ to achieve the desired therapeutic effect, the patient should have his/her blood glucose and cholesterol levels checked in addition to a physical check of the heart function (pulse, blood pressure, and EKG). To be on the safe side, a complete blood picture (CBP) and a survey of hepatorenal functions is also usually requested.


**Switching therapy to another antipsychotic agent:**


The switching issue should only be conducted under supervision of a psychiatrist. This is indicated usually in two conditions: the first is due to inadequate efficacy of a certain APC and the second is unacceptable adverse effects of the agent, such as weight gain, of which occurs in the case of second-generation APCs [57,58]. We say a drug has inadequate efficacy when

(1)The drug could not neutralize positive or negative symptoms of psychosis;(2)Psychosis relapses despite the patient’s cooperation (medication adherence) and treatment trial for typically 4–6 weeks in a sufficient therapeutic dose;(3)Functional disabilities and poor control of chronic symptoms are observed;(4)Mood symptoms and cognitive dysfunction respond better to the other intended drug to be switched to.

Unacceptable adverse effects, such as in the case of OLZ, and significant metabolic side effects are the best known reasons for switching. Other reasons include weight gain above 5% of the baseline body weight, high cholesterol, or type 2 diabetes [59,60].

Other side effects that might necessitate switching to another alternative (alternatives) are EPS, persistent sedation, persistent cognitive problems, sexual dysfunction, increased QT interval on EKG, and tardive dyskinesia.


**Benefit to risk profile when switching antipsychotic agents:**


The best known intervention is what is called plateau cross-titration. The strategy constitutes slower titration of the new agent while keeping the first agent in its current therapeutic dose [61]. This is a sort of overlapping strategy between the first drug and the other drug intended for the switch. This method secures maintaining the first drug at its current dose, while gradually initiating and increasing the new drug to an effective therapeutic dose, followed by a gradual tapering of the first agent.

The advantages of this strategy are as follows: minimizes relapse risk, manages withdrawal symptoms that might emerge, and allows close monitoring of the adverse effects from drugs of both the new and the old medication.

The disadvantages of the strategy are as follows: additive side effects (two drugs), drug interactions (of the two agents), patient’s adherence to the therapy resultant of an increase in the number of pills, and long duration of switch. Patient education and participation in decision making is crucial. There are other strategies to switch from one drug to a new one, but they are indeed less favorable and fall outside the scope of this review work.

## 2. Discussion

The metabolic drawbacks of OLZ were the main reason for the rigorous investigation of this lipophilic second generation APC, especially its pharmacodynamics. Otherwise, this agent is potent against the negative symptoms of schizophrenia (which are difficult to treat and the real problems of schizophrenic individuals). The drug is also one of the alternatives to treat affective disorders (although as an add-on) in patients with depressive symptoms, in addition to its many important off-label uses. The drug regimen for each case should be individualized depending on clinical outcome and blood concentration [62].

**Overweight:** Many agents, including drugs and hormone groups, are known to cause patients to be overweight or experience obesity [63,64,65]. Among oral hypoglycemic drugs, sulfonylurea agents can induce weight gain indirectly via increasing endogenous insulin levels [64,66]. Thiazolidinedione type oral hypoglycemic agents such as pioglitazone and rosiglitazone cause fluid retention, thus promoting adipogenesis through the activation of peroxisome proliferator-activated receptor gamma (PPARγ) [67,68,69]. Antihypertensive drugs, of beta-blocker type, e.g., propranolol and atenolol, are among the medicines that can cause weight gain via different mechanisms [70,71,72] explained in Table 2. The other class of antihypertensive agents is calcium channel blockers. In this case, weight gain of the body is linked to its blocking effects on both calcium and dopamine receptors [73,74]. The mood stabilizer lithium can also induce weight gain in some individuals, although this is a controversial issue. It causes increased thirst and changes in food preference. Lithium also influences thyroid function and increases the incidence of hypothyroidism, which is another cause of weight gain [75,76]. The atypical APC Sulpiride blocks dopamine D2 receptor of the lateral hypothalamus which is involved in satiety. It also blocks pituitary D2 receptors. The latter condition results in hyperprolactinemia, creating a condition similar to ovariectomy which in turn induces hyperphagia and weight gain [64,66,77]. Tricyclic antidepressants, e.g., amitriptyline and nortriptyline, block different subclasses of histamine receptors. These interfere with serotonin reuptake, which controls appetite and increases the craving for carbohydrate-rich diet. Moreover, they cause hypoglycemia by increasing the circulating insulin amount, thus inducing insulin resistance [78,79]. Serotonin-inhibiting antidepressants include selective serotonin reuptake inhibitors (SSRIs) and serotonin–norepinephrine reuptake inhibitors (SNRIs), and the agents of these two antidepressant groups indeed cause a slight weight loss to start with but with prolonged therapy, many agents of these two groups cause weight gain [79,80]. Glucocorticoids increase craving for high-calorie diet carbohydrates and fat, through changes in the activity of AMP-activated protein kinase in the hypothalamus [81] (see Table 2). Anticonvulsants such as valproate, carbamazepine, pregabalin, and gabapentin have a considerable impact in this parameter of side effects [82,83], and the mechanisms are also explained in Table 2.

Psychotropics (antipsychotic agents) have a big impact on this issue [84]. Indeed, one of the most significant drawbacks of second generation APCs (especially OLZ) is weight gain. This assertion is also true even when OLZ is used to treat off-label conditions [85]. It is worth mentioning that different agents of second generation APCs have different categories of metabolic side effects [86].The medical literature includes dozens of documents to confirm this undesired pharmacodynamic effect of OLZ. The multiple treatment meta-analysis conducted by Leucht et al. using six-weeks of data elucidated that most of the currently available APCs, both first and second generation agents, can cause weight gain; the exception is with haloperidol, lurasidone, and ziprasidone. Among these, OLZ and clozapine caused the highest (significant) weight gain, while quetiapine, risperidone, and sertindole caused milder effects according to a head-to-head meta-analysis [87]. An Indian study confirms that 66.6% of 80 patients who received OLZ had a gain weight of 1–5 kg over a period of 4 weeks; this was neither dependent on OLZ dose being administered nor BMI (body mass index) of these individuals being included in the study. The other significant finding in this study is as follows: the weight gain was related to females of 40 years of age and upwards [88]. Females over 40 years of age are more prone to OLZ-induced weight gain than younger females and men of any age [50,88]. Women are probably more susceptible to olanzapine-induced weight gain over men for different reasons. Indeed, weight gain attributed to OLZ therapy is a serious drawback psychiatrists should consider when deciding to prescribe OLZ to their patients, at least to those who already suffer from metabolic conditions such as hyperlipidemia and/or insulin resistance [27]. OLZ affects almost all classes (and their subclasses) of human membrane receptors [89] and among these are the histamine receptor subclasses (H1, H2, H3, and H4). Antagonism of these receptors by OLZ causes weight gain [90]. According to a meta-analysis conducted on 12,279 patients of 40 randomized and 15 uncontrolled trials, it is possible to reverse weight gain to a certain extent, but not completely to baseline weight. The measures that have been applied to reverse the weight gain included drug discontinuation, dose reduction, switching to monotherapy, or switching to a partial agonist agent [56]. Switching from OLZ to asenapine might reverse OLZ-induced obesity to a certain degree, according to a case report, but unfortunately, this agent also has its own drawbacks, such as sedation and somnolence. Sometimes an adjunct add-on such as aripiprazole on OLZ might be beneficial according to recent systemic review work [91]. Switching of medication might be an option, but one has to take into consideration the relapse of the illness [55]. It is best that the tapering of OLZ should be intermingled synchronously with the dose escalation of the alternative APC agent which is planned for the switch. The other significant factor that causes weight gain in schizophrenic individuals using OLZ as therapy is that the drug, per se, acts as an appetizer; some individuals taking neuroleptics report craving for sweet and fatty food [92]. The regulation of appetite in humans is controlled by the hypothalamus. This physiological process is sophisticated and not well elucidated. The action of the hypothalamus is integrated with the signals received from other parts of the brain and via hormones released from the adipose tissue and the gastrointestinal tract. The most significant signal transducer hormones in this context are leptin and ghrelin [84]. Leptin regulates NPY and POMC neurons of the lateral hypothalamic area (arcuate nucleus). These neurons regulate food intake and body weight. NYP neurons are orexigenic because they stimulate appetite; on the other hand, POMC neurons are anorexigenic, which means they suppress appetite [93]. OLZ upregulates the expression of NPY while downregulating the expression of the POMC pathway in the hypothalamus. This imbalance enhances food intake and causes weight gain [94]. There might be other parameters that can influence the pharmacodynamic effects of OLZ but they have yet to be elucidated.

It is believed that certain neurotransmitter receptors play a role in this complex process. Evidence indicates the implication of serotonin receptors (5-HT2C and 5-HT1A), such as histamine H1 and dopamine D2, in addition to many other receptors. APC agents differ in their potency to block these receptors, which is a fact that may explain their different propensities to induce weight gain. OLZ and clozapine antagonize the action of these receptors; thus, these agents have a higher risk of causing individuals to be overweight [95] (see Table 3). The ratio of weight gain also varies among patients who use these agents, which is a fact that implies differences between individuals regarding their dietary habits, level of daily physical activity, and their genetic makeup. Polymorphism in the gene that codes 5-HT2C has been blamed mostly in this context, in addition to other genes [96,97]. Recent research has elucidated the contribution of serotonin-signaling in the regulation of glucose homeostasis via serotonylation of certain proteins known as Rab4 [98]. Another issue that should be noted here is that the selective antagonism of 5-HT2 (mainly by second generation APC agents, which include OLZ) impairs insulin sensitivity. These nonconventional APCs, moreover, induce abnormal differentiation of adipocytes [99], increase the accumulation of lipids in hepatocytes [100], upregulate the synthesis of sterol regulatory element-binding protein [101], and inhibit the accumulation of glycogen in skeletal myocytes [102]. Despite these proposed mechanisms to explain the induction of insulin resistance (consequently, the weight gains), the issue of weight gain is indeed a matter of debate and controversy among clinicians, endocrinologists, and drug designers.

It has to be admitted that currently available data in the medical literature regarding the molecular mechanisms behind the toxicity exerted by second generation APCs is limited, and cannot provide a plausible explanation for the wide-spectrum side effect profile these agents exhibit as a pharmacodynamic parameter. In fact, in vivo studies in rodents have shown that OLZ impairs insulin sensitivity in hepatocytes [103], skeletal myocytes, and adipocytes [104,105,106]. Additionally, recent research has demonstrated that OLZ reduces insulin-mediated glucose uptake through mechanisms involving impaired hypothalamic insulin-sensing, as shown by pancreatic glycemic clamp studies [106]. Overall, in vitro research suggests that OLZ can induce insulin resistance in tissues throughout the body [100,101,102].

We do not know precisely how second generation agents operate at the cellular level and this is attributed to the fact that we do not know the pathophysiology and the biochemical changes that occur in neurons of psychotic and, to be more precise, schizophrenic individuals; it is not OLZ alone that should be blamed. Polymorphism in HTR2C receptors has been proved to be an important factor in OLZ-induced weight gain [107]. HTR2C polymorphism has variations in the human receptor gene htr2c which encodes serotonin 2C receptor. This receptor has a pivotal role in the regulation of serotonin-signaling in the brain. The significance of these variations is linked to the etiology of many psychiatric diseases, including psychoses; additionally, they can also influence the pharmacodynamic effects of these drugs, including adverse effects such as weight gain and other metabolic adverse effects [108,109], as well as the therapeutic response to APC agents, which constitutes both positive and negative symptoms of schizophrenia, especially polymorphism (−759 C/T) [110]. Animal studies also indicate that 5-HT2C plays a crucial role in regulating appetite and energy homeostasis. If the gene that encodes this receptor protein is removed, as demonstrated in knockout mice, they show hyperphagia, insulin resistance, and obesity [111,112]. People with schizophrenia are more liable to gain weight (almost twice as much as individuals in the general population) because of APC therapy and other adjunct drugs that are frequently needed in combination to achieve the most benefit from APC therapy, or because of coexisting symptoms (signs) that might be difficult to control with a single psychotropic agent. The other aspect of the schizophrenia spectrum is isolation from society and physical inactivity because of psychosis and its consequences, even in those who are drug naïve. In fact, individuals who are on APC therapy consider the weight gain as the most fundamental drawback of these agents. Moreover, we have to realize that being overweight increases the risk of developing heart attack, stroke, and many other physical illnesses such as hypertension, insulin resistance (thus type 2 diabetes), osteoarthrosis, and sleep apnea [113]. All these consequential morbidities enhance physical inactivity and make things much worse. It is an accepted fact that the maximal APC-induced weight gain occurs within the first 6 months of the therapy period, especially in the first month, which is an indicator for greater weight gain in the long run, then the rate declines [114]. The weight gain is more prominent in individuals who use APCs for the first time (acute episode of psychosis) than in those who have to start the same medication again. This could be attributed to the relapse of symptoms after discontinuation or because of uncooperative adherence to the same APC for the same disease or other mental condition that necessitates the reiteration of the same APC. Bear in mind that we are talking about a 7% increase or more of the starting weight [115].

Clinicians believe there is no clear link between APC dose and weight gain [114]. Fortunately, many agents within what is known as the second generation drugs are available. Clinicians can choose among these drugs, which generally provide similar therapeutic effects but differ in their likelihood to cause weight gain [84,114], as demonstrated in Table 3. The exception is clozapine, which is ideal for treating negative symptoms of schizophrenia (such as avolition, anhedonia, and inability to act spontaneously), as well as resistant schizophrenia and mania, despite its serious potential side effects [30]. The cellular mechanisms behind weight gain are not fully understood, but it is well-known that this agent can cause hyperglycemia (insulin resistance) and hyperlipidemia through different mechanisms discussed in this review. According to the US FDA–1440 × 810, it is believed that combining OLZ with samidorphan (Lybalvi) reduces the risk of weight gain compared to OLZ alone in monotherapy [116,117,118]. Additionally, metformin, an oral hypoglycemic agent, has been used as an adjunct to OLZ to prevent weight gain associated with this and other second generation APCs that may have metabolic side effects [119]. The 5HT2C receptor agonist lorcaserin can suppress or even reverse OLZ-induced weight gain, by activating this receptor through indirectly antagonizing the action of OLZ. Lorcaserin, namely, activates the 5HT2C receptor on anorexigenic neurons of the hypothalamus, thus promoting the feeling of satiety to decrease food intake [120].


**Insulin resistance:**


The reason why OLZ causes obesity, or at least increases weight, is attributed to the predisposition of insulin resistance [21,121,122,123]. Insulin resistance is a state of prediabetes where glucose transporters GLUTs (mainly GLUT1, GLUT2, GLUT3, and GLUT4) do not respond to the action of the hormone insulin on insulin receptors (a defect in the signal transduction cascade initiated by insulin). The increase in insulin levels in the circulation causes lipogenesis within cells, especially in lipocytes, and consequently leads to weight gain [124]. The two main known contributing factors to insulin resistance are excess body fat around the waist and lack of physical activity. The etiology of insulin resistance has not been elucidated hitherto, but it is generally accepted that the underlying causes are either genetic or acquired. What concerns us in this context is the acquired factors, namely, the drug-induced ones. There are many drug groups known to cause insulin resistance, which include steroids, certain antihypertensive drugs, some antivirals that are used in HIV treatments, and certain neuroleptics APC agents (see Table 3). One has to admit that insulin resistance is a complex morbidity because the condition does not exhibit prominent symptoms until it progresses into prediabetes or type 2 diabetes. But once it occurs, the best-known scientific methods to prevent/reverse the condition is via changing dietary habits to healthy ones, and regular physical exercising to keep the body healthy and lean, which is a condition unfortunately not easily achieved in psychotic patients.

Indeed, insulin resistance, especially in drug-induced cases, is a complicated subject of debate in endocrinology and diabetology. This is attributed to the fact that its exact mechanism and pathophysiology is not elucidated convincingly. Genetic factors, including mitochondrial dysfunction, have been blamed as being among the most important contributing factors of inducing this prediabetic metabolic disturbance resulting from resistance to the function of insulin [125,126,127,128]. In other words, insulin resistance can occur in schizophrenia even before the administration of APC agents; therefore, many clinicians and scientists classify schizophrenia under mitochondrial diseases [126]. Results obtained from both preclinical and clinical studies have shown that among the seond generation APC agents, OLZ is the strongest in terms of its ability to induce metabolic toxicity via its effects on plasma glucose levels, weight gain, and many other metabolic parameters [129]. Serotonin improves insulin sensitivity and glucose metabolism, but how it does this is elusive. It is believed that serotonin activates certain intracellular small GPTases known as (rab4) proteins, by covalently binding to these proteins in a process known as sertonylation. These serotonylated proteins are associated with increased translocation of GLUT4 glucose transporters to the cell membrane [98]. To neutralize symptoms of psychosis in schizophrenia, OLZ antagonizes the serotonin receptor, thus hindering serotonin action on its receptors, impairing insulin sensitivity, and causing weight gain. This assertion is backed by administering of a selective antagonist [130].

Despite their wide-spectrum metabolic adverse effects, clozapine, risperidone, OLZ, quetiapine, and aripiprazole have been among the top-selling pharmaceuticals worldwide over the last decade [131,132].

**Mitochondrial dysfunction:** For decades, mitochondrial dysfunction has been recognized as a potential predisposing factor to both psychosis and affective disorders [126,133]. Intact and sound mitochondria in neurons is a must for energy production (oxidative phosphorylation) and Ca^++^ homeostasis, which are crucial for the action of neurons. A deviation in mitochondria-related intracellular biological pathways that regulate neuronal survival, apoptosis, oxidative stress, plasticity, and neuronal transmission has been observed both in psychosis and affective disorders [89,125,134,135,136,137,138]. In other words, malfunctioning neuronal mitochondria distort the connectivity of neuronal networks, which might be the reason behind the abnormal emotional and cognitive behaviors that manifest clinically both in schizophrenia and affective disorders. In this connection, in an animal model, transplantation of intact healthy mitochondria into neurons has improved the aberrant mitochondrial function and restored the behavioral deficits in animals [139]. Mitochondrial dysfunction and serotonin receptor signaling are interconnected, which means mitochondrial dysfunction disrupts serotonin-signaling and serotonin receptors can regulate mitochondrial function. Stimulating specific serotonin receptors can enhance improvement in mitochondrial health in certain diseases, while defects in mitochondrial function can lead to impairment in serotonin release and metabolism. This intermingled relationship suggests using the targeting of serotonin-signaling to treat diseases related to dysfunctioning mitochondria, including neuropsychiatric diseases [140].

The interesting issue here is that mitochondria are a target for psychotropic agents, which contribute to the modulatory effects on mitochondrial gene expression and thus the synthesis of proteins that are fundamental for the mitochondrial machinery to drive citric acid cycle, oxidative phosphorylation, apoptosis, autophagy, and other dynamics of the mitochondrial network [89,95,141,142,143,144]. OLZ releases free radicals in neurons and thus causes oxidative stress; accordingly, it induces depolarization of mitochondrial membranes and subsequently their damage, thus causing neuronal apoptosis as neurons cannot survive without intact and sound mitochondria to synthesize ATP [145]. Regarding the release of free radicals, one has to admit that the second generation agent OLZ has a milder propensity than the first generation agents (such as haloperidol [146]) and even other second generation agents (such as clozapine [30] and quetiapine [147]) when releasing reactive oxygen species. As psychotic individuals respond differently to different APC agents, recent studies apply mitochondrial function as a parameter to choose the best currently available APC for each psychotic patient separately [148]. Still, neither the first nor the second generation agents are optimal. The reason for this is that we know very little about the biochemical changes that stand behind the pathophysiology of psychosis [30].


**Is olanzapine addictive?**


Although it shares features that mimic other agents that can cause abuse, such as benzodiazepines, Z-drugs (nonbenzodiazepines, used to treat insomnia), and barbiturates, there exists no clear and strong evidence based on a reliable systematic study to demonstrate that OLZ is an addictive agent. One of the arguments that backs the assumption that OLZ may not possess the high potential to cause abuse is that the agent has a long elimination half-life (t/2) in humans (about 33 h) [40]. Moreover, abusive drugs usually have a shorter duration of action, as is the case with amphetamine, cocaine, etc. Yet clinicians should be cautious when prescribing this thienobenzodiazepine derivative to individuals who have records of substance abuse. Bear in mind that this substance is water-insoluble and readily passes the blood barrier in a manner that mimics that of diazepam, and both of these agents have CNS depressant properties, which is why it is expected that OLZ could be misused as a recreational drug.


**Tapering off olanzapine and withdrawal symptoms of olanzapine cutoff:**


Similar to other psychotropic agents, such as APCs and antidepressants, OLZ should be tapered off gradually under the supervision of a psychiatrist. This is attributed to the fact that abrupt cutoff of OLZ can cause rebound symptoms, which means relapse of the symptoms of psychosis that the drug was originally prescribed to treat [45,149]. The other serious issue here is the emergence of withdrawal symptoms [150,151,152]. These should be thoroughly monitored within the tapering period. They can vary both in severity and duration and manifest as follows:(A)Psychological symptoms: Anxiety, agitation, irritability, restlessness, insomnia and mood disturbances.(B)Physical symptoms: Nausea, vomiting diarrhea, headaches, dizziness, sweating, tremors, myalgia, and abnormal skin sensations.(C)Withdrawal dyskinesia: Involuntary contraction of the face and other muscles of the body, causing bizarre movements of the body.(D)Autonomic symptoms: Tachycardia, hot or cold flashes.

If withdrawal symptoms start to emerge, the psychiatric caregiver may recommend returning to the previous tolerated therapeutic dose and starting the tapering process again but this should be performed more slowly.


**Monitoring of relapse of underlying psychotic condition:**


This risk is high, especially if the tapering was performed quickly. The psychotic symptoms could mimic withdrawal symptoms and are difficult to distinguish. The milestones of rebound psychosis may emerge as follows:(1)Psychotic symptoms: Delusions, hallucinations, paranoid reactions, and hostility (mistrust).(2)Mood symptoms: Depression, mania (or mixture of both), especially in those prescribed with APCs during treatment.(3)Increased distress and suicidal thoughts: Close and regular follow-up with a psychiatrist is essential for at least 3–6 months after completely stopping the medication to assess the possibility of symptom relapse.

Regular health check-ups should continue during and after the cutoff process; this is because OLZ affects various bodily functions. The check-up should include the following: weight (BMI), fasting blood glucose, blood lipid levels, prolactin concentration (whenever symptoms of hyperprolactinemia such as breast enlargement, galactorrhea appear), and signs of NMS [153].

When the decision to stop the medication is made, certain issues have to be assessed: (1) The reason for discontinuation: Is it lack of efficacy, side effect of the agent, or patient preference? (2) Stability of the psychiatric condition: Remission of symptoms of acute psychosis, etc. (3) Is OLZ interacting with other drugs the patient is using that cannot be stopped? When the decision is to discontinue with OLZ, the tapering schedule depends on the dose of OLZ the patient is using.

A: For standard doses (5–20 mg/day), the rule here is to reduce the dose by 25% every 3–4 weeks. This means that if the daily dose is 20 mg, the plan would be as follows: week 1–4: 20 mg/day; week 5–8: 15 mg/day; week 9–12: 11.25 mg/day; week 13–16: 8.5 mg/day; week 17–20: 6.5 mg/day; and week 37: discontinuation.

B: For higher doses, more than (20 mg/day), the dose should be reduced to the standard dose (20 mg/day) by using 25% reduction in the high dose [154]. Then, one can follow the same previous rule applied for standard doses, that is 25% reduction for each quartile (each 3 weeks).

The evidence-based hyperbolic tapering rationale for APC discontinuation mentioned above may trigger dopamine hypersensitivity syndrome (DSS) which can persist for months after stopping the medication [39].

## 3. Conclusions

In this work, we have discussed the plausible and possible mechanisms behind insulin resistance and overweight in individuals who are on OLZ therapy. We have attempted to assemble the pieces of the puzzle to understand the spectrum of adverse metabolic effects associated with this agent. We believe that considerable effort should be made to fill the gaps in our understanding of the etiology of the unwanted metabolic effects that OLZ exerts. In fact, OLZ is one of the best APCs we have to combat the negative symptoms of schizophrenia. Monitoring of weight and the other accompanying metabolic parameters is necessary before starting OLZ therapy. This is necessary in cases when we have to switch to another agent (or other agents when a combination therapy is in question) to obtain the desired therapeutic response. A lot more research has to be conducted to understand the real biochemical changes that occur in neurons (and the mitochondria they possess) of schizophrenic individuals. This is important as certain agents of second generation APCs, such as OLZ, are lipophilic molecules that can easily cross the blood–brain barrier and cellular and organelle membranes, such as mitochondria (the energy house of neurons). As schizophrenic individuals already have dysfunctional mitochondria in their neurons, the future (optimal) APC agents should devoid the property of free radical release that can jeopardize mitochondrial function and aggravate psychotic symptoms. Moreover, they should be more selective in such a way that they only antagonize the receptors they are designed to block, because OLZ and almost all currently available psychotropic agents affect almost all classes of neuronal membrane receptors to various degrees. Clinicians have many alternative agents within second generation APCs that they can choose to prescribe for each patient, which is a condition that necessitates cooperation between the individual patients and the psychiatrist. The prescribed APC should be fitted to the individual patient, because patients respond variably to different agents; in other words, what is convenient for one patient does not necessarily fit the other one for therapy. The future perspectives should be focused on designing better drugs that are selective and potent in their therapeutic action, while having a narrow profile of adverse effects.

## Figures and Tables

**Figure 1 jcm-14-08125-f001:**
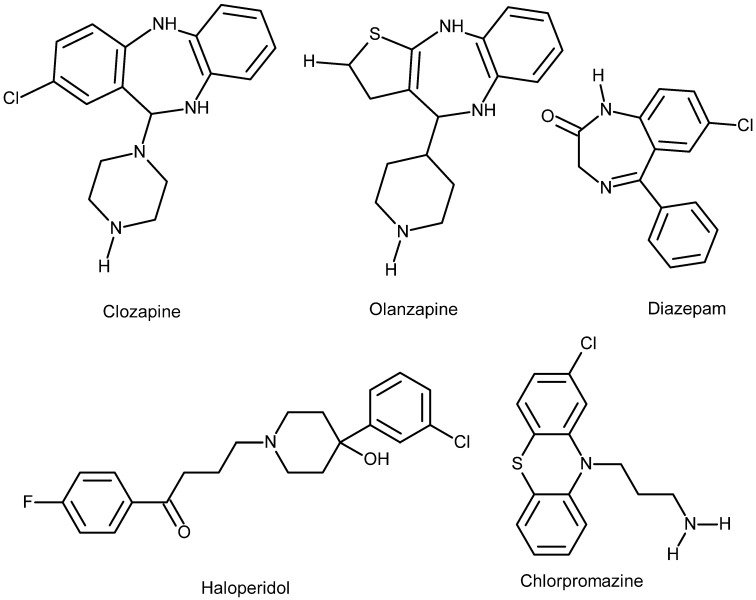
**Chemical structure of the drugs discussed in this work, illustrating their solubility properties. Diazepam, Olanzapine, and Clozapine** are nearly insoluble in water, slightly soluble in alcohol, acetone, and dichloromethane, and highly soluble in chloroform because they are lipophilic molecules. **Haloperidol** is slightly soluble in water, soluble in alcohol, and freely soluble in chloroform. **Chlorpromazine** (demonstrated for comparison) is an amphiphilic molecule, meaning it is soluble in both water and organic solvents. Data from PubChem.

**Table 1 jcm-14-08125-t001:** Metabolic monitoring protocol for adults on second-generation antipsychotic drugs.

	Baseline	1 Month	2 Months	3 Months	6 Months	Reassessment
Weight (BMI) *	X	X	X	X	X	Q 3 months
Waist circumference	X	X	X	X	X	Q 3 months
Blood pressure	X			X	X	Q 3 months for 1 year, then annually
Fasting blood sugar	X			X	X	Q 3 months for 1 year, then annually
Fasting lipid profile	X			X		Annually

Adapted from the recommendation of American Diabetes Association and American Psychiatric Association; a cross reference in reference [54]. BMI * = body mass index (weight Kg/height m^2^).

**Table 2 jcm-14-08125-t002:** Agents (drugs/hormones) that induce weight gain, with references.

SN	Type of Drug	Examples	Mechanism	References
**1.**	**Anti-diabetic drugs**			
		**Insulin (hormone)**	Circulating insulin is an anabolic hormone. In type 2 diabetes, the induced weight gain is due to a reduction in the signaling of satiety to the arcuate nucleus of the hypothalamus.	[63,64,65]
		**Sulfonylurea agents**, such as glyburide, glipizide, and glimepiride	These increase endogenous insulin levels.	[64,66]
		**Thiazolidinediones,** such as pioglitazone and rosiglitazone	They induce weight gain due to fluid retention, the promotion of lipid storage, and adipogenesis through the activation of peroxisome proliferator-activated receptor gamma (PPARγ).	[67,68,69]
**2.**	**Antihypertensive drugs**			
		**Beta-blockers, such as** propranolol and atenolol	These affect body weight through two main mechanisms: (**1**) reductions in total energy expenditure through lowering of the basal metabolic rate and thermogenic response to meals, and (**2**) inhibition of lipolysis in response to adrenergic stimulation. Moreover, these agents can promote fatigue and reductions in patient activity.	[70,71,72]
		**Calcium channel blockers,** such as Flunarizine	Body weight gain is linked to its blocking effects on calcium channels and dopamine receptors.	[73,74]
**3.**	**Drugs acting on CNS**			
		**Antipsychotics**	Olanzapine, clozapine, chlorpromazine, quetiapine, risperidone, and paliperidone.	[84]
		**Anticonvulsants,** such as Valproate, carbamazepine, pregabalin, and gabapentin	Valproate causes weight gain through (**1**) Central mechanism: Via interactions with appetite-regulating neuropeptides and cytokines within the hypothalamus, as well as effects on energy expenditure; (**2**) Peripheral actions: Perturbation of glucose and lipid metabolism that contribute to weight-independent worsening of insulin resistance and the risk for type 2 diabetes.	[73,82]
		**Mood stabilizers:** lithium	The possible mechanisms include (**1**) direct effect on hypothalamic centers that control appetite, increased thirst, increased intake of high-calorie drinks, and changes in food preference, and (**2**) its influence on thyroid function with increased incidence of hypothyroidism.	[75,76]
		**Sulpiride**	It blocks (**1**) D2 dopamine receptors in the lateral hypothalamus which are involved in satiety. (**2**) It also blocks the pituitary D2 receptors involved in the inhibition of prolactin secretion, which results in hyperprolactinemia, creating a condition similar to a functional ovariectomy, which in turn induces hyperphagia and weight gain.	[66,77]
	Antidepressants	**TCA**: Amitriptyline and Nortriptyline	(**1**) Block different classes of histamine receptors. (**2**) Interfere with the reuptake of serotonin, which controls appetite, and increases craving for carbohydrate-rich food. (**3**) They cause hypoglycemia by increasing circulating blood insulin and inducing insulin resistance.	[78,79]
		**Serotonin agents:** (**1**) SSRIs, such as citalopram, fluoxetine, and sertraline. (**2**) SNRIs, such as venlafaxine and duloxetine	Indeed, these are associated with a slight weight loss to start with, but with prolonged therapy, many of these agents have been shown to cause weight gain in individuals who use them for treatment.	[79,80]
H4.	Endocrine agents			
		Glucocorticoids	These may induce an increase in food intake and dietary preference for high-calorie, high-fat (comfort foods) foods through changes in The activity of AMP-activated protein kinase in the hypothalamus.	[81]

**Table 3 jcm-14-08125-t003:** Antipsychotic agents that can cause weight gain, sequenced in increasing order, and their affinity to different receptor groups.

Antipsychotic Agent	Weight Gain Risk	Ki Value, nM/L		
		htr2c	h1	m3
Haloperidol	Low	4.700	3.000	1000
Ziprasidone	Low	0.72	47	negligible
Lurasidone	Low	415	1000	>1000
Aripiprazole	Low	15	61	1.500
Amisulpride	Low	Very low	>10.000	>10.000
Asenapine	Low	0.034	1.0	>1000
Paliperidone	Medium	178	3.4–34	>10.000
Risperidone	Medium	50	2.23–15	10.000
Quetiapine Norquetiapine *	Medium	600–1800	4.41–10	1320 10–100 *
Chlorpromazine	Medium/high	27.1	4.25	47
Clozapine	High	9.4	1.1–2	11 ± 1.0
Olanzapine	High	6.4–29	7.0	13–132

This table is adapted from the guidelines of the British Association of Psychopharmacology (reference [77], Drug Bank, and DFA report). * The active metabolite of quetiapine (norquetiapine) has a different Ki value to the m3 receptor than quetiapine. The drugs are listed in order of their risk of causing weight gain from lowest to highest, with their Ki values to different receptor classes.

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
