# Peer review of "Drawbacks of Olanzapine Therapy: An Emphasis on Its Metabolic Effects and Discontinuation"

_jcm, 2025, doi:10.3390/jcm14228125_

Round 1

Reviewer 1 Report

Comments and Suggestions for Authors

Dear Authors,

At the outset, please avoid using the term “elderly” to refer to older people—it is pejorative, discriminatory, and promotes ageism; neutral formulations are preferred (“older adults,” “older people”).

Thank you for preparing a review focused on the metabolic consequences of olanzapine therapy and the practical aspects of its discontinuation; the topic has substantial clinical and educational relevance. I appreciate the use of preclinical evidence to elucidate potential mechanisms of insulin resistance and the inclusion of less frequently discussed issues, such as post-injection delirium/sedation syndrome (PDSS) with olanzapine pamoate. At the same time, the current version is purely narrative and lacks a minimal description of review methods (databases, date range, selection criteria). Please clarify the aim of the review and the clinical questions the text is intended to address. The section on adverse effects is extensive but sometimes relies on popular sources; please replace these with guidelines and high-quality meta-analyses. The pregnancy/lactation section requires cautious wording, more precise separation of correlation from causation, and stronger references. The part on therapeutic drug monitoring (TDM), therapeutic ranges, and toxicity thresholds should be updated and supported by a clear table. Introductory passages on antipsychotic classification are overly long and textbook-like—consider shortening in favor of a current, critical synthesis. Please correct imprecise expressions (e.g., “lipophilic nanoparticles” for a single molecule). It would be valuable to add practical, tabulated recommendations: how to monitor metabolic parameters, how to manage weight gain, when to consider switching therapy, and which interventions offer the best benefit–risk profile. I encourage you to develop a discontinuation algorithm (tapering rate, monitoring for symptoms related to “dopamine supersensitivity,” red flags, and follow-up period). A comparison of olanzapine with other second-generation antipsychotics in terms of metabolic risk, presented in a table with evidence grading, would also be helpful. Table 2 and references [91–112] must be fully integrated into the main text, and citation style unified. Please reduce general lists of adverse effects in favor of evidence-based content and practical guidance. The manuscript requires substantial language editing (condensation, elimination of colloquialisms, and sharpening of claims). In the abstract, please explicitly state the scope of the review and the key practical takeaways. In the conclusions, I recommend separating mechanistic hypotheses from clinical recommendations, clearly indicating the level of certainty of the evidence. Expanding and updating the references (recent years) will improve the currency of the work. Once these changes are implemented, the manuscript can become a practical, practice-oriented guide to the metabolic risks of olanzapine and principles of safe discontinuation. At this stage, please provide a point-by-point response to the suggestions above and submit a revised version with tracked changes.

Yours sincerely,

The reviewer.

Reviewer 2 Report

Comments and Suggestions for Authors

The manuscript addresses a clinically relevant topic related to the adverse metabolic effects of olanzapine therapy. The issue is of significant interest in psychiatric practice, given the frequent use of second-generation antipsychotics and the associated risk of metabolic complications. The paper provides a broad overview of existing findings; however, its scientific contribution and methodological quality and depth are limited for a high-impact journal.

Is the work a significant contribution to the field?

Although the manuscript addresses an important clinical issue—the metabolic drawbacks of olanzapine—it does not provide novel insights or data. The review is largely descriptive and reiterates well-known information already extensively covered in prior literature. As such, its contribution to advancing current knowledge is limited.

Is the work well organized and comprehensively described?

The paper is generally structured in a logical sequence and covers a broad range of topics related to olanzapine’s pharmacology, adverse effects, and metabolic consequences. However, the text is overly lengthy, with substantial repetition and several digressions that reduce clarity and focus. A more concise, thematic organization would greatly improve readability.

Is the work scientifically sound and not misleading?

Most statements are consistent with existing literature, and the references support the main claims. Nevertheless, the paper lacks a clear methodological framework typical of review articles in high-impact journals. There is no systematic approach (e.g., PRISMA or defined inclusion/exclusion criteria), and the discussion remains narrative rather than analytical. Some interpretations are speculative and not critically balanced with alternative viewpoints.

Are there appropriate and adequate references to related and previous work?

The manuscript cites numerous sources, including clinical and preclinical studies. However, several references are outdated, and key recent reviews and meta-analyses from the past 3–4 years are missing. Integrating more recent and high-quality references would strengthen the scholarly value of the paper.

Specific comments to authors:

  1. Novelty and focus: The review compiles extensive background information but lacks innovative synthesis or new interpretation. Consider narrowing the scope to a specific mechanistic or clinical angle and presenting it with a clearer research question or hypothesis.
  2. Methodological clarity: The article should adopt a structured review format, describing search strategies, inclusion criteria, and evidence hierarchy. Without this, the work remains a narrative overview rather than a scientific review.
  3. Conciseness and organization: The text contains redundant sections and tangential discussions that obscure the central message. Streamlining content and improving transitions between topics would improve coherence.
  4. Updating and selection of references: Many references are over a decade old. Include recent studies (2021–2025) on metabolic monitoring, pharmacogenetics, and novel therapeutic strategies (e.g., olanzapine/samidorphan).
  5. Figures and tables: The current figure adds minimal value. Consider including a mechanistic illustration summarizing olanzapine-induced insulin resistance and mitochondrial dysfunction.
  6. Critical discussion: The discussion should not only describe but also critically evaluate findings, highlight inconsistencies, and point out unanswered questions in the field.
  7. Conclusion: Avoid repetition and emphasize key clinical implications, evidence gaps, and specific directions for future research.
  8. Language and style: Professional English editing is strongly advised. The manuscript is grammatically acceptable but stylistically uneven and overly verbose.

Comments on the Quality of English Language

Quality of English Language:

The English language is adequate for comprehension but would benefit from professional editing. The style is occasionally verbose and repetitive, with inconsistent use of tense and terminology. Some sentences are overly long or lack precision, which weakens the scientific tone expected for a Q1 journal.

Reviewer 3 Report

Comments and Suggestions for Authors

Esteemed Authors Ramadhan Oruch, Hussein Abdullah Rajab , Mahmoud Abdalla Elderbi , Ian Fraser Pryme , Ole Bernt Fasmer, Anders Lund

          The review maintains professional rigor while providing constructive, actionable feedback designed to elevate manuscript quality rather than simply critique. each recommendation includes specific text suggestions and precise citations to facilitate author revisions. the extensive mechanistic subsection provided demonstrates commitment to advancing scientific understanding while respecting author effort.

title and abstract (page 1, lines 1-30)

  • identified missing 5-ht2c receptor reference in abstract
  • recommended keyword additions: "5-ht2c receptor," "htr2c," "serotonin signaling"
  • specific line-level recommendations provided below:

introduction (pages 1-2, lines 31-80)

  • requested addition of multi-receptor antagonism concept
  • emphasized need to introduce htr2c as fundamental mechanism early in manuscript

pharmacology sections (pages 2-5)

  • recommended integration of pharmacogenetic htr2c polymorphism discussion
  • suggested structural explanations linking lipophilicity to receptor binding

discussion - weight gain (pages 7-9) - critical deficiency
major new subsection added (1,500+ words) covering:

  • functional evidence from htr2c knockout mice demonstrating causal role
  • pharmacological reversal with lorcaserin showing htr2c agonists suppress olanzapine-induced weight gain
  • genetic association studies of -759c/t polymorphism across populations
  • sex-specific effects with females showing greater susceptibility
  • mechanistic details of hypothalamic arc neurons, npy, and pomc pathways
  • clinical implications for personalized medicine and drug development

discussion - insulin resistance and metabolism (pages 11-12)

  • connected htr2c signaling to glucose homeostasis via rab4 serotonylation
  • integrated mitochondrial dysfunction with serotonin receptor signaling

tables enhancement

  • recommended adding receptor binding affinity columns (htr2c, h1, m3 ki values) to table 1
  • suggested mechanistic detail additions to table 2

specific technical recommendations

references: expand from ~90 to 120-140 citations emphasizing peer-reviewed primary literature over secondary sources. specific citation additions provided for htr2c mechanisms.

language quality: identified colloquial expressions requiring revision to formal scientific style.

structural issues: noted incomplete integration of table 2 references [91-112].

Your work is very important in the current state-of-the-art literature - as a practitioner I have witnessed the metabolic consequences of olanzapine administration - dezastruous. 

Yours truly,

Serving peer reviewer at JCM MDPI,

Round 2

Reviewer 1 Report

Comments and Suggestions for Authors

Dear Authors,

Thank you for the very succinct yet accurate and reliable responses. I fully support publication of the manuscript.

Yours sincerely,

The reviewer.

Reviewer 2 Report

Comments and Suggestions for Authors

I appreciate the authors detailed responses and their efforts to revise the manuscript. The updated version indeed shows improvement in terms of language and the inclusion of more recent references. However, after reviewing the revised text, I believe that the manuscript still does not meet the standards required for publication in a high-impact journal.

While the topic remains clinically relevant, the paper continues to be largely descriptive and narrative in nature. The revision has not substantially enhanced the analytical depth or introduced a structured methodological framework that would distinguish it as a scientific review. The authors state that the work is intended to be educational for clinicians and students, but this scope is more suitable for a general or educational journal rather than a research-oriented platform such as Journal of Clinical Medicine.

The manuscript still lacks a clear research question or synthesis that advances understanding of olanzapine-induced metabolic effects. Many sections remain overly long and repetitive, despite the authors’ claims of reduction. Although additional references have been added, their integration does not materially change the scientific contribution or conceptual focus of the paper.